# Olfactory bulb acetylcholine release dishabituates odor responses and reinstates odor investigation

M. Cameron Ogg[1], Jordan M. Ross[1], Mounir Bendahmane[1] & Max L. Fletcher[1]

Habituation and dishabituation modulate the neural resources and behavioral significance allocated to incoming stimuli across the sensory systems. We characterize these processes in the mouse olfactory bulb (OB) and uncover a role for OB acetylcholine (ACh) in physiological and behavioral olfactory dishabituation. We use calcium imaging in both awake and anesthetized mice to determine the time course and magnitude of OB glomerular habituation during a prolonged odor presentation. In addition, we develop a novel behavioral investigation paradigm to determine how prolonged odor input affects odor salience. We find that manipulating OB ACh release during prolonged odor presentations using electrical or optogenetic stimulation rapidly modulates habituated glomerular odor responses and odor salience, causing mice to suddenly investigate a previously ignored odor. To demonstrate the ethological validity of this effect, we show that changing the visual context can lead to dishabituation of odor investigation behavior, which is blocked by cholinergic antagonists in the OB.

---

[1] Department of Anatomy and Neurobiology, University of Tennessee Health Science Center, Memphis, TN 38163, USA. Correspondence and requests for materials should be addressed to M.L.F. (email: mfletch4@uthsc.edu)

Neuronal and behavioral responses to prolonged stimuli decrease over time as sensory systems habituate to non-relevant stimuli. Yet, if unexpected changes within an environment occur, reestablishing sensory responsivity to previously filtered stimuli can be beneficial for survival. Though these processes, habituation and dishabituation, are ubiquitous across the sensory systems, the mechanisms through which they are mediated are often complex and vary depending on the system[1]. The aim of this study was to investigate these two important sensory processing phenomena in the mouse olfactory bulb (OB).

The OBs form the first part of the olfactory central nervous system, where sensory information from the nasal epithelium is processed before projecting to cortical areas. The axons of peripheral olfactory sensory neurons (OSNs) synapse onto glomeruli, dense collections of the dendrites of OB output (mitral/tufted; M/T) cells and interneurons. Odor information contained in OSN activity is transformed into spatiotemporal patterns of glomerular responses that represent both odor quality and intensity[2–4].

In a previous study using in vivo calcium imaging, we found that a single, prolonged odor pulse decreases subsequent M/T cell glomerular odor responses for minutes following the initial presentation[5]. However, it is currently unknown the extent to which these responses are reduced during continuous odor presentation. Using calcium imaging and a novel behavioral paradigm, we examined habituation of glomerular odor responses and odor investigation behavior during prolonged odor presentations. In addition, we wanted to determine if, once habituated, glomerular odor responses could be reinstated, what effect this would have on behavioral odor investigation, and whether a cholinergic mechanism could be responsible.

The cholinergic system plays an important role in olfactory learning, processing, and perception[6–9], however, except for a few studies focused on olfactory discrimination[10, 11], its role in the OB in modulating olfactory behaviors remains largely unexplored. The OB receives cholinergic input from the basal forebrain (BF) and expresses a variety of cholinergic receptors[12–16]. Recent work from our lab and others has demonstrated that OB acetylcholine (ACh) release increases sensitivity to odor input at both the glomerular and M/T cell output levels[17–19], mediated by muscarinic receptor activation[17]. This, together with growing evidence of rapid, phasic ACh modulation in other brain regions[20], led us to hypothesize that brief ACh release in the OB could dishabituate glomerular responses during prolonged odors and, as a result, allow the stimuli to be detected and investigated again.

To test this cholinergic dishabituation hypothesis, we manipulated OB ACh release electrically and optogenetically during prolonged odor presentations. We found that ACh can rapidly modulate habituated excitatory post-synaptic glomerular odor response and increase odor salience. Further, we determined that this change in odor investigation behavior happens naturally in response to contextual changes in the environment, and can be blocked using a cholinergic antagonist in the OB.

## Results

**Glomerular responses to prolonged odor presentation**. To determine if prolonged odor input leads to decreased or habituated post-synaptic glomerular odor responses, we imaged glomerular odor responses in anesthetized Thy1-GCaMP3 mice before, during, and after 30-s odor presentations (Fig. 1a). Overall, we found glomerular responses display a rapid rise in response amplitude that quickly reaches a maximum and is followed by a slower, steady decay to approximately 40% of the peak response by end of the odor presentation ($n = 92$ glomeruli from 6 mice; paired $t$-test: $t(91) = 32.15$, $p = <0.0001$; end = 36.74 ±

1.97%; Fig. 1b, c). We did not observe any glomerular responses that maintained maximal responses or increased in amplitude over the course of the odor presentation.

Based on previous work from our lab demonstrating that OB muscarinic ACh receptor (mAChR) activation enhances glomerular odor responses[17], we hypothesized that brief OB ACh release delivered near the end of the odor presentation could reinstate, or dishabituate, reduced OB odor responses. To test whether ACh release could enhance OB responsivity to adapted stimuli, we again imaged glomerular odor responses in anesthetized Thy1-GCaMP3 mice implanted with a stimulating electrode in the cholinergic BF[17]. For these experiments, BF electrical stimulation (BFS) was delivered 24 s into the 30 s odor stimulation (Fig. 2a). As in the control mice, mean glomerular odor responses decreased during the odor presentation. BFS delivered at 50 Hz rapidly increased these habituated glomerular odor responses (Fig. 2b, c). To quantify this, we compared the mean normalized fluorescence in the second immediately preceding the BFS to the mean normalized fluorescence in the second following BFS for each glomerulus and found a significant increase in the mean fluorescence following BFS ($n = 76$ glomeruli from 5 mice; repeated measures anaysis of variance (RM ANOVA): $F(2,150) = 749.8$, $p < 0.0001$; pre-BFS = 35.31 ± 1.13%, post-BFS = 78.64 ± 1.62%; Fig. 2d).

In a subset of mice, in addition to 50 Hz stimulation we also stimulated at 5 Hz, a frequency previously shown to have little effect on OB activity[18] (Fig. 2b, c). This allowed us to directly compare response enhancement at two different stimulation frequencies in the same mice. A two-way RM ANOVA revealed a significant interaction between imaging time and BFS stimulation ($n = 32$ glomeruli from 2 mice; $F(1,62) = 1015.43$, $p < 0.0001$). Post hoc tests revealed significant increases when BFS was given at 50 Hz (pre-BFS = 32.24 ± 1.51%, post-BFS = 88.00 ± 2.33%), but no significant changes in fluorescence at 5 Hz (pre-BFS = 30.14 ± 1.38%, post-BFS = 30.15 ± 1.51%; Fig. 2e). Prior work from our lab demonstrated that BFS-driven glomerular enhancement was due to OB muscarinic receptor activation, as the effect could be completely blocked with the muscarinic antagonist scopolamine[17]. To confirm our dishabituation effect was mediated by the same receptors, we repeated the dishabituation paradigm in additional two mice in which scopolamine (100 µM) was applied to the OB through as small slit in the dura. We found a significant reduction in the mean BFS-induced fluorescence increase when scopolamine was applied compared to controls (paired $t$-test: $t(21) = 6.56$, $p = 0.0001$; control $\Delta F = 109.5 ± 16.2$, Scop $\Delta F = 26.3 ± 5.6$).

To investigate these effects in awake, behaving animals, we developed a simple olfactory habituation paradigm to measure mice's natural odor investigation behavior. In this paradigm, the duration of stereotypical investigation behaviors was scored and quantified (Supplementary Fig. 1a, b). To demonstrate that the duration of these behaviors is an appropriate measurement of olfactory salience in the mouse, we compared the change in the duration of odor investigation from baseline during clean air ($n = 8$ mice) and odor (ethyl butyrate, 10% dilution; $n = 8$ mice) presentations (Fig. 3a, b, Supplementary Fig. 1b). Odor investigation behavior was measured for 5 min, with clean air or odor presented for the third minute only. For the clean air group, we found no differences in investigation time per minute (RM ANOVA: $F(4,28) = 2.39$, $p = 0.07$; minute 1 = 18.2 ± 1.7 s, 2 = 13.2 ± 1.6 s, 3 = 18.2 ± 1.5 s, 4 = 16.0 ± 1.9 s, and 5 = 18.3 ± 2.2 s; Fig. 3a).

There was a significant effect of odor on investigation time (RM ANOVA $F(4,28) = 15.73$, $p < 0.0001$; minute 1 = 15.1 ± 1.8 s, 2 = 14.8 ± 1.1 s, 3 = 30.0 ± 1.6 s, 4 = 22.7 ± 2.3 s, and 5 = 16.9 ± 1.9 s; Fig. 3b). The post hoc test revealed that investigation time in

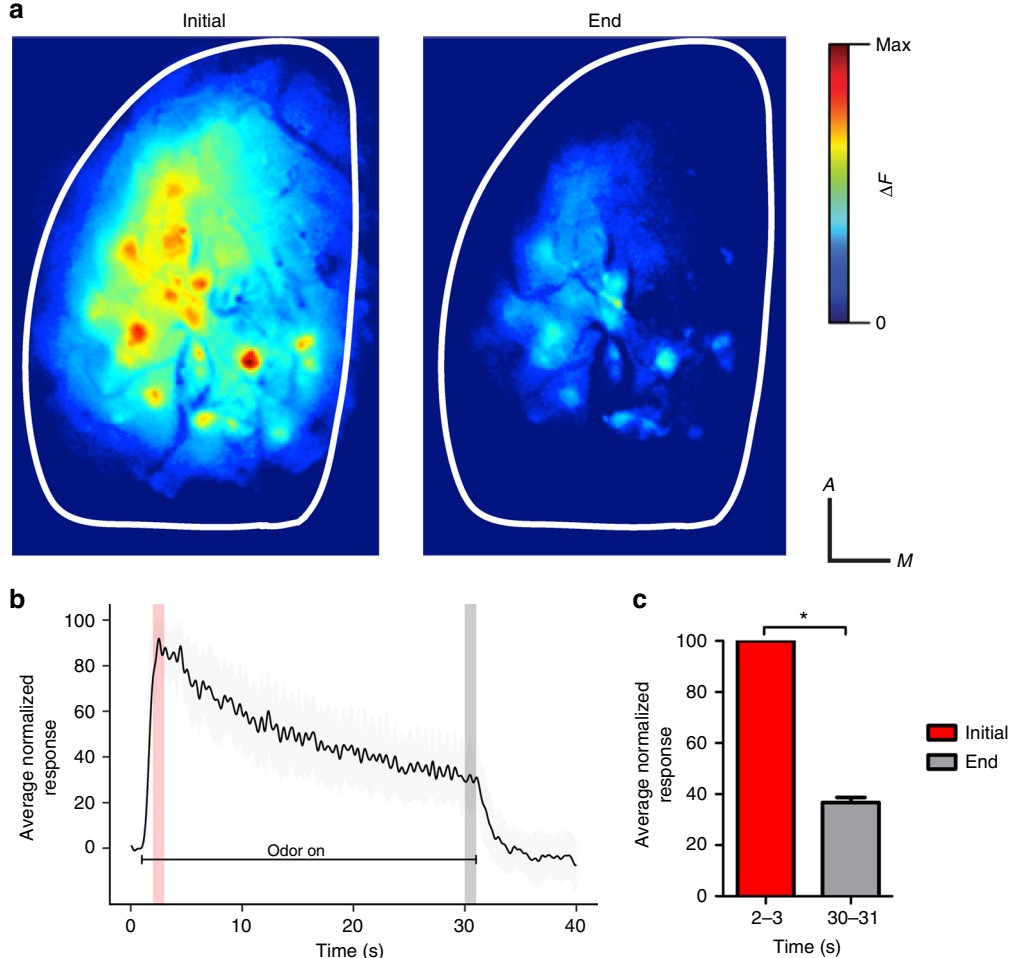

**Fig. 1** Glomerular responses to a prolonged odor presentation habituate over time in anesthetized mice. **a** Pseudo-color glomerular responses to 2-heptanone (0.3% s.v.) initially and after 30 s of the odor presentation. **b** The average fluorescence trace (% of initial response ± s.e.m.) from all recorded glomeruli ($n = 96$, 6 mice). **c** The average glomerular odor response (% of initial response ± s.e.m) decreases during the odor presentation. *$p < 0.05$

the third minute when odor was present was significantly higher than in any other non-odor minute. To demonstrate that this paradigm accurately reflects odor investigation and that investigation times are not dependent upon other factors, we also compared investigation times for 1 min presentations of different odors as well as the to the same odor across 2 days. No differences were seen in odor-evoked investigation times regardless of odor or day (Supplementary Information, Supplementary Fig. 1d, e).

Next, we sought to determine if our olfactory investigation paradigm could be used for measuring olfactory habituation. We quantified investigation behavior during one baseline minute (minute 0) and 6 min of continuous odor (ethyl butyrate) and found significant differences in investigation time across minutes ($n = 10$; RM ANOVA: $F(6,9) = 17.95$, $p < 0.0001$; minute $0 = 16.66 ± 1.67$ s, $1 = 32.50 ± 1.59$ s, and $6 = 16.77 ± 1.91$ s; Fig. 3c). The post hoc test revealed that odor investigation increased significantly from baseline when an odor was delivered into the chamber (minute 0 vs 1). Further, when an odor was continuously delivered into the chamber for several minutes, odor investigation significantly decreased (minute 1 vs 6) demonstrating that this task can effectively measure olfactory habituation.

To directly investigate whether increased ACh release in the OB can reinstate investigation of habituated odors in awake, behaving animals, we used mice expressing channelrhodopsin in cholinergic neurons (ChAT-ChR2) and their wild-type (WT)

littermates in the above the habituation paradigm. In this case, odor (isoamyl acetate) was presented for 9 min. No effect of genotype was found between WT ($n = 5$; minute $0 = 13.22 ± 1.46$, $1 = 32.16 ± 1.96$, and $6 = 7.72 ± 1.88$) and ChR2+ ($n = 5$; minute $0 = 11.77 ± 2.88$ s, $1 = 25.68 ± 3.06$ s, and $6 = 11.38 ± 1.01$ s) mice when comparing investigation duration at key time points in the habituation paradigm (two-way RM ANOVA: $F(1,8) = 0.43$, $p = 0.53$). To specifically drive optogenetic OB ACh release, we turned on a light-emitting diode (LED) implanted over the OBs[21] (optogenetic light stimulation, OLS; Fig. 4a). OLS (3 s, 50 Hz) at the beginning of the seventh minute of odor exposure, following habituation, did not increase odor investigation during the seventh minute compared to the previous minute in WT mice (minute $6 = 7.72 ± 1.88$ and $7 = 9.80 ± 1.98$; Fig. 4b). However, in ChR2+ mice, investigation time after the 50 Hz OLS increased back to the level of initial odor investigation, and stayed increased through the end of the trial (RM ANOVA: $F(9,36) = 4.57$, $p = 0.0005$; minute $6 = 11.38 ± 1.01$ s, $7 = 28.04 ± 2.56$ s, and $9 = 19.92 ± 3.88$ s; Fig. 4c–e).

As found in a previous study[18] and in our imaging data above, there is an intensity-dependent effect of cholinergic activation and OB response enhancement. Therefore, we repeated the experiment in the same ChR2+ mice with two additional stimulation frequencies (5 and 25 Hz). Decreasing the stimulation frequency reduced the extent and duration of subsequent investigation. After 25 Hz OLS, investigation time increased back

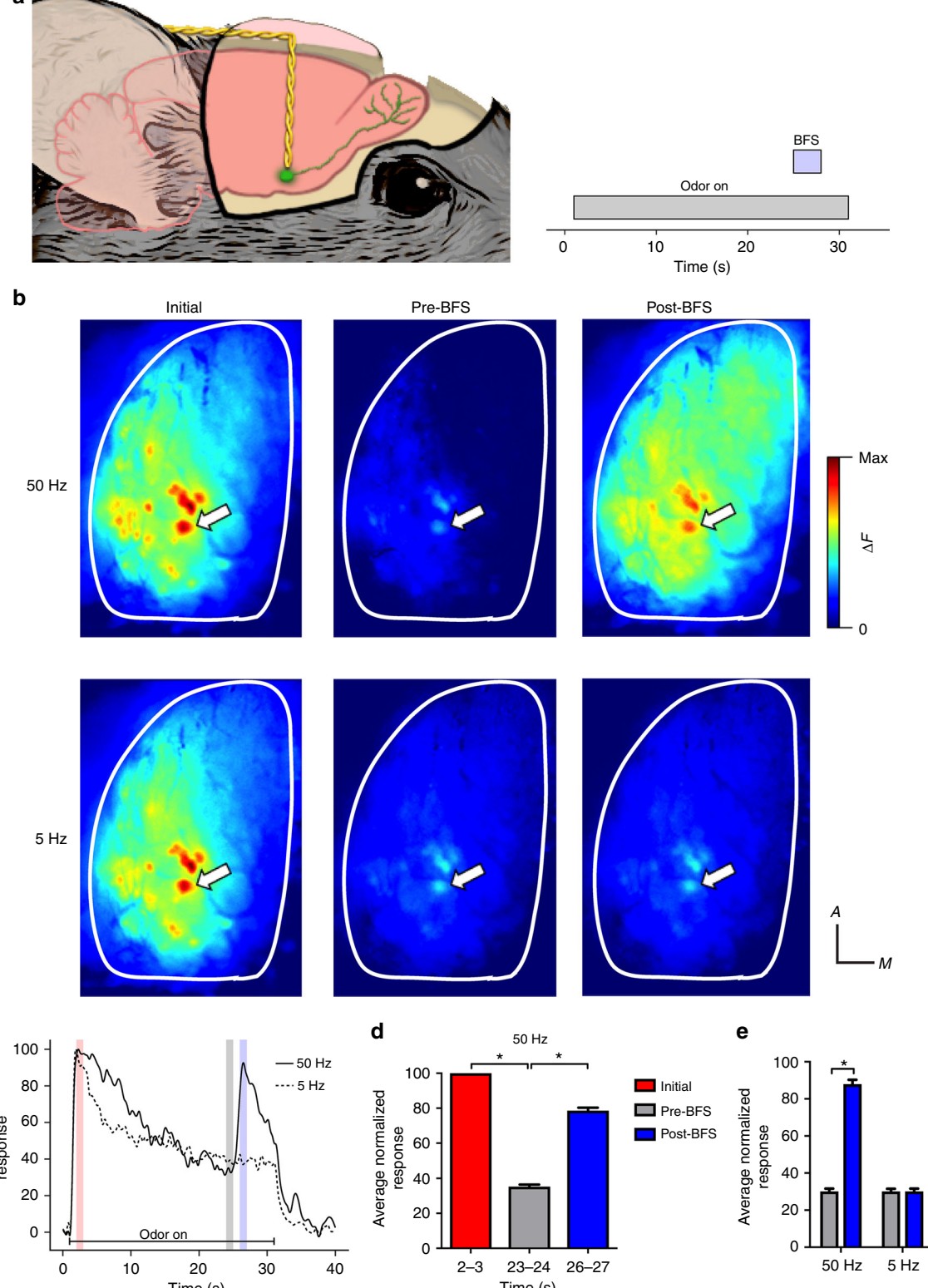

**Fig. 2** Electrical basal forebrain stimulation (BFS) dishabituates glomerular odor responses in anesthetized mice. **a** Schematic illustration of the imaging experiment. **b** Pseudo-color glomerular responses to 2-heptanone (0.5% s.v.) initially and before and after BFS (50 μA, 3 s) at 50 and 5 Hz. **c** Fluorescence traces (% of initial response) from the glomerulus indicated by arrows in **b**. **d** The average glomerular odor response (% of initial response ± s.e.m) decreases during the odor presentation and increases following 50 Hz BFS (n = 76, 5 mice). **e** In a subset of mice, 5 Hz stimulation was given in addition to 50 Hz, and it did not increase the average glomerular odor response. *p < 0.05. Illustration by M.C.O.

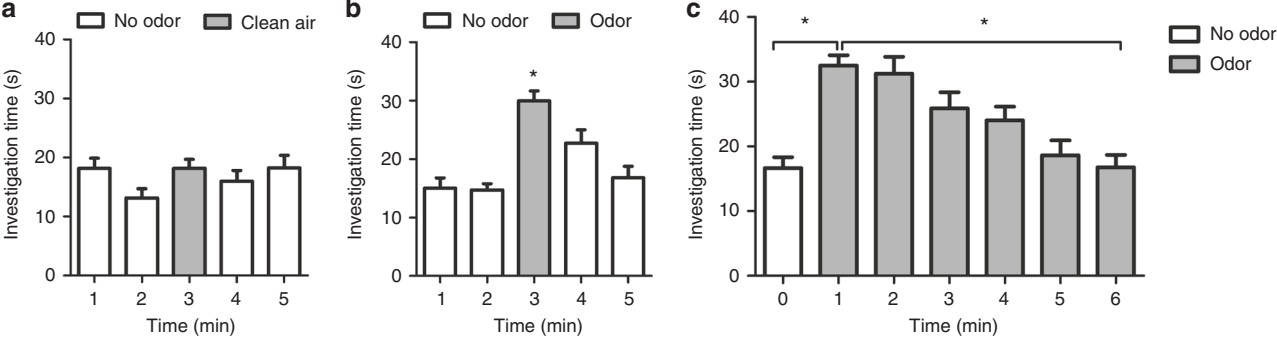

**Fig. 3** Olfactory investigation behavior increases with odor onset and habituates over time. **a** Investigation time (seconds) is not affected when odor is not present, but **b** increases with odor (10% ethyl butyrate) onset ($n = 10$ mice). **c** Investigation time in control mice increases with odor onset and decreases over the exposure ($n = 10$ mice). Error bars = s.e.m., *$p < 0.05$

to the level of initial odor investigation, but did not remain increased (RM ANOVA: $F_{(9,36)} = 12.02$, $p = < 0.0001$; minute 0 $= 9.95 \pm 2.38$ s, 1 $= 28.46 \pm 4.23$ s, 6 $= 13.08 \pm 1.80$ s, 7 $= 23.66 \pm 3.07$ s, and 9 $= 10.24 \pm 2.79$ s) and 5 Hz OLS did not increase investigation time at all (RM ANOVA: $F_{(9,36)} = 4.50$, $p = 0.0005$; minute 0 $= 12.77 \pm 1.77$ s, 1 $= 23.50 \pm 2.08$ s, 6 $= 11.60 \pm 2.34$ s, 7 $= 11.42 \pm 1.64$ s, and 9 $= 13.94 \pm 2.49$ s; Fig. 4d, e).

Previous studies have demonstrated that cholinergic activation alters M/T cell spontaneous activity[18, 19]. As potential changes in spontaneous M/T cell activity could potentially affect olfactory driven behavior, we repeated the experiment in an additional cohort of ChR2+ ($n = 5$) mice in the absence of any odor presentation and found no changes in investigation after optogenetic stimulation (paired $t$-test: $t_{(4)} = 1.17$, $p = 0.31$; minute 0 $= 13.48 \pm 2.76$ s, 1 $= 10.52 \pm 3.06$ s; Fig. 4e, f). Further, in an additional set of three mice we monitored sniffing throughout the odor presentation and OLS. Sniff rates did not change following OLS onset suggesting that OLS does not in of itself lead to increased sniffing (Supplementary Fig. 2).

While the LED studies demonstrate that ACh fiber activation within the OB is sufficient to driven dishabituation behaviorally, we next tested whether this effect can occur under more naturalistic conditions. As previous studies have demonstrated novel sensory stimuli lead to ACh release across cortex and hippocampus[22, 23], we hypothesized that a novel change in visual context could drive ACh release across the brain, including the OB. This increased OB ACh would act on AChRs to dishabituate OB odor responses similar to OLS-driven release. To test this, we performed the habituation experiment again, but instead of artificially driving ACh release at the beginning of the seventh minute of odor exposure, we abruptly changed the visual context of the chamber (VS; Fig. 5a). Based on our findings here and a previous imaging study from our lab showing that mAChR activation in the OB leads to increased odor responses[17], we further hypothesized that blocking mAChR activation would prevent any contextual change driven olfactory dishabituation. Mice ($n = 10$) were implanted with bilateral OB cannula and received infusions of the mAChR antagonist scopolamine (1 mM) prior to one trial and infusions of vehicle (Ringer's solution) prior to another. Mice were counter balanced so that half received vehicle first and the other half received scopolamine first. No effect of treatment was found when comparing investigation duration at key time points (minutes 0, 1, and 6) in the habituation paradigm (two-way RM ANOVA: $F_{(1,18)} = 1.86$, $p = 0.19$). Further, both scopolamine- and vehicle-treated mice displayed similar habituation rates with investigation times first becoming significantly different than the initial minute by minute three.

Following vehicle infusion, investigation behavior increased with odor (ethyl valerate) onset and decreased throughout the odor presentation (RM ANOVA: $F_{(9,81)} = 10.71$, $p = 0.0001$; minute 0 $= 9.55 \pm 1.14$ s, 1 $= 29.37 \pm 2.38$ s, and 6 $= 11.34 \pm 1.90$ s). Similar to the optogenetic ACh stimulation, changing the visual context in the seventh minute of the odor presentation significantly increased investigation behavior for the next three minutes (minute 6 $= 11.34 \pm 1.90$ s, 7 $= 18.75 \pm 2.39$ s, and 9 $= 13.21 \pm 2.43$ s; Fig. 5b, c). When the same mice were treated with OB scopolamine, investigation still significantly increased with the initial odor presentation and decreased by minute 6 (RM ANOVA: $F_{(9,81)} = 14.26$, $p = 0.0001$; minute 0 $= 8.96 \pm 0.96$ s, 1 $= 24.19 \pm 2.09$ s, and 6 $= 9.36 \pm 1.94$ s), but the visual stimulation-mediated increase in odor investigation was completely blocked (minute 6 $= 9.36 \pm 1.94$ s, 7 $= 6.87 \pm 1.19$ s, and 9 $= 11.14 \pm 1.50$ s; Fig. 5c, d), indicating this form of dishabituation requires OB mAChR activation. To verify that changing the visual context alone does not increase odor investigation behaviors or that the effects seen are not due to changes in general arousal, we also repeated experiment in non-cannulated mice ($n = 3$) in the absence of odor and found no changes in investigation after the context switch (paired $t$-test: $t_{(2)} = 0.09$, $p = 0.93$; minute 6 $= 14.47 \pm 2.67$ s and 7 $= 14.90 \pm 2.13$ s, Fig. 5d).

As OB odor responses[24–26] and cholinergic modulation of mitral-granule cell interactions[27] can be quite different between anesthetized and awake states, we repeated our imaging experiments in awake, head-fixed mice. These mice received either 30 s odor trials, as in the anesthetized experiments, or 7 min odor trials, as in the behavioral paradigm. In both cases control mice displayed significant decreases in glomerular response during the trial (30 s: $n = 96$ glomeruli from 4 mice; paired $t$-test: $t_{(95)} = 55.63$, $p = < 0.0001$; end $= 11.92 \pm 15.51\%$; Fig. 6a, b, control; 7 min: $n = 92$ glomeruli from 4 mice; paired $t$-test: $t_{(91)} = 7.54$, $p = < 0.0001$; end $= 22.56 \pm 14.72\%$; Fig. 6c, control). We compared the habituation amount at the last second of the 30 s odor presentation in awake mice to that of anesthetized mice and found that habituated responses were significantly lower in awake mice ($t$-test: $t_{(186)} = 11.91$, $p < 0.0001$; anesthetized: $36.74 \pm 1.97$, awake: $9.66 \pm 1.83$).

At either 25 s or 6 min, a black and white lined image identical to the one used in the behavioral paradigm was projected onto a screen in front the mouse. Using post hoc tests, we compared the mean normalized fluorescence second before and after the visual stimulus for each glomerulus and found a significant increase in the mean fluorescence following the visual stimulation for both the 30 s ($n = 73$ glomeruli from 4 mice; RM ANOVA: $F_{(2,144)} = 612.1$, $p < 0.0001$; pre-VS $= 11.52 \pm 3.30\%$, post-VS $= 34.96 \pm 2.58\%$; Fig. 6a, b, visual stimulation) and 7 min exposures

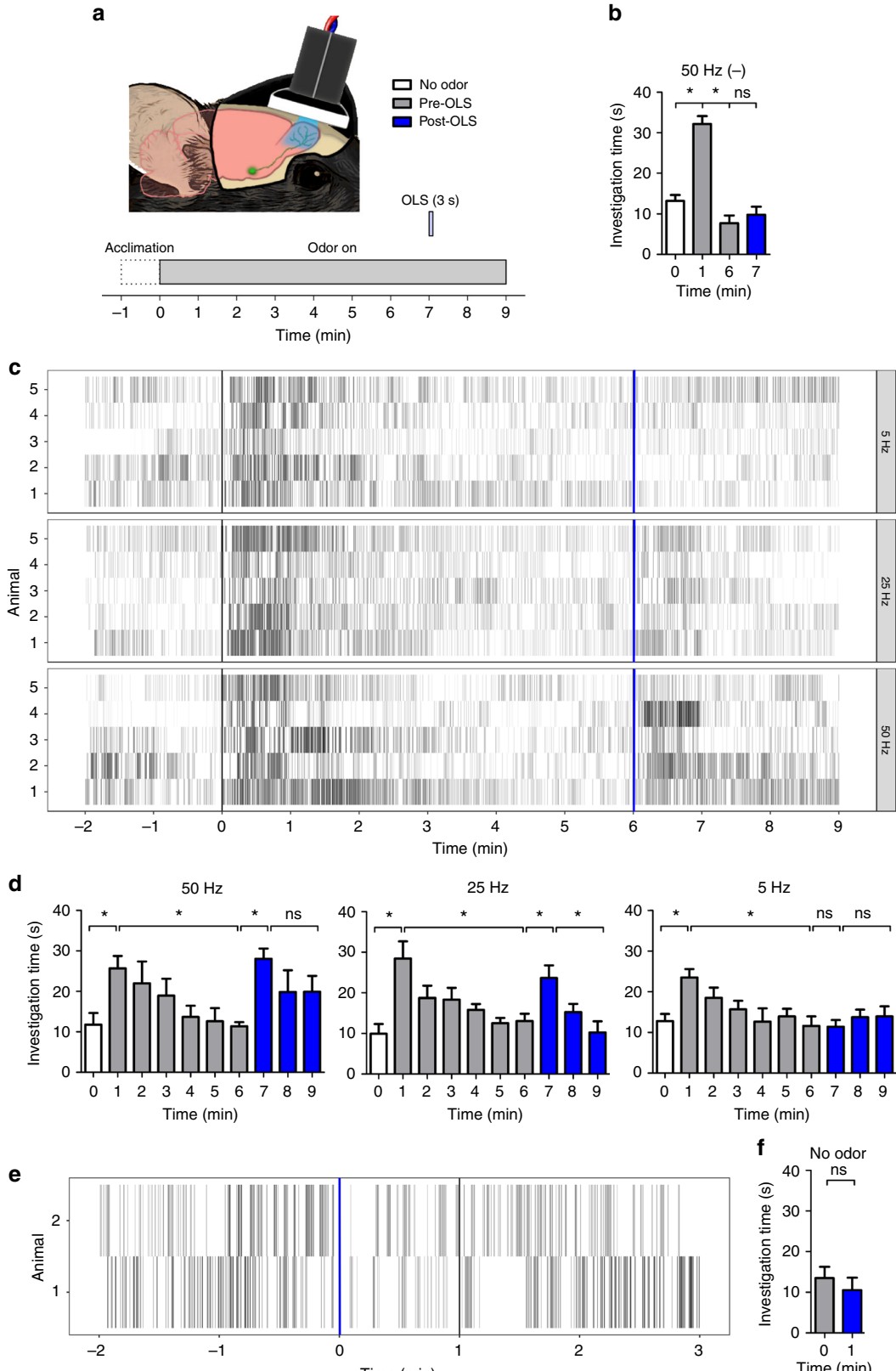

**Fig. 4** Optogenetic light stimulation (OLS) can dishabituate odor investigation behavior. **a** Schematic illustration of the OLS experiment. **b** Investigation time (seconds ± s.e.m.) does not increase after 50 Hz OLS in WT (ChR2−) controls (*n* = 5 mice). **c** Raster plots of the odor investigation behavior comparing OLS at 5, 25, and 50 Hz in the same mice (*n* = 5 mice). **d** Investigation time in Ch2R+ mice after the OLS is stimulation dependent: following 50 Hz OLS, investigation time increases and stays increased for at least 3 min. After 25 HZ OLS, investigation time also increases, but does not remain increased. 5 Hz OLS does not increase investigation time. **e** Raster plots of investigation behavior taken from two mice before and after OLS (blue line) in the absence of any odor stimuli demonstrating the OLS itself does not lead to increased sniffing. **f** Mean investigation times (seconds ± s.e.m.) taken from the minute before and after OLS in mice in the absence odor stimulation. *\*p* < 0.05. Illustration by Ogg, M.C.

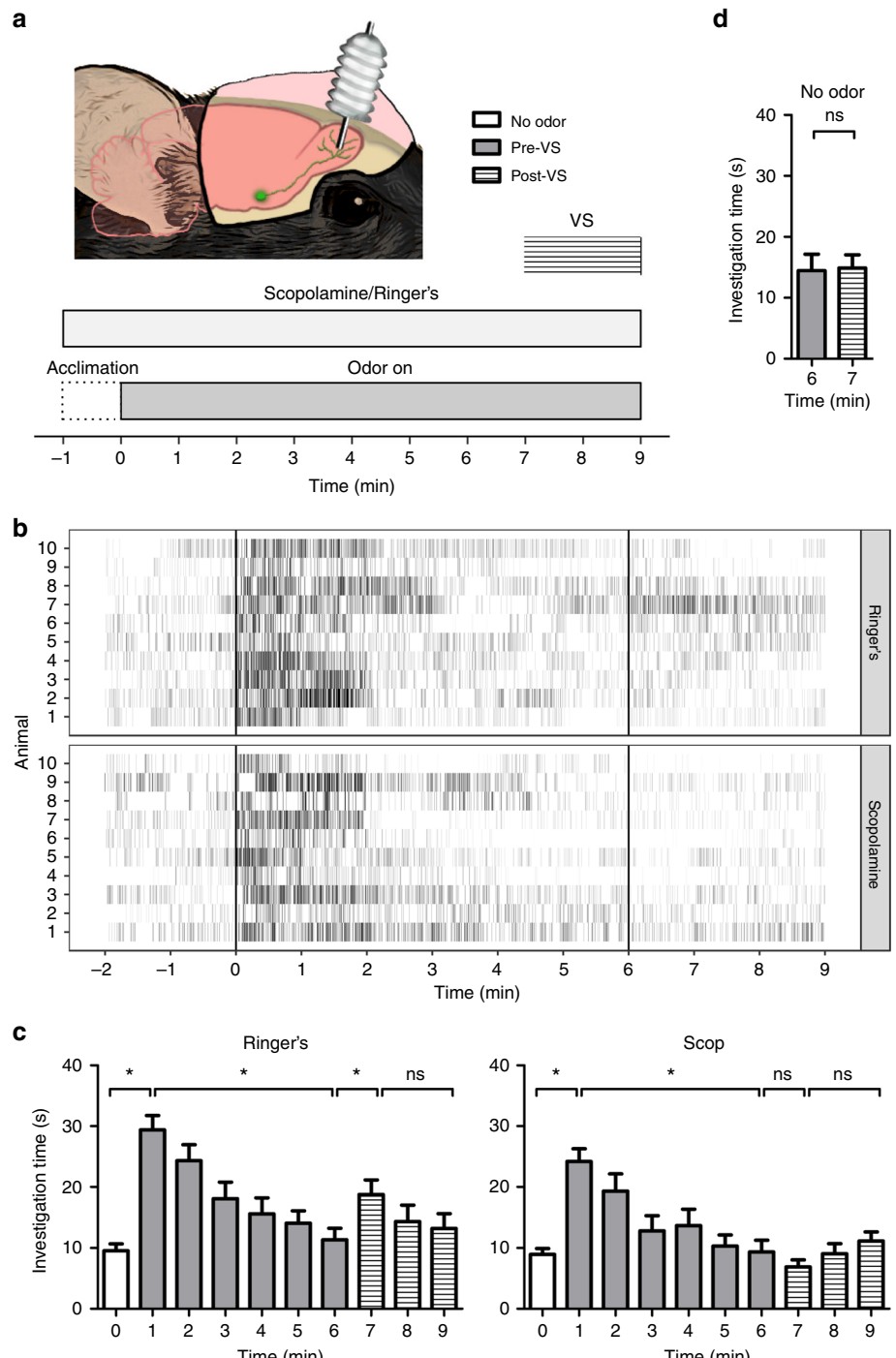

**Fig. 5** A visual context change (VS) dishabituates odor investigation behavior. **a** Schematic illustration of the VS experiment. **b** Investigation time (seconds ± s.e.m.) after the VS does not increase if odor is not present ($n = 5$ mice). **c** Raster plots of the odor investigation behavior following olfactory bulb (OB) cannula infusion of vehicle (Ringer's) or the cholinergic antagonist scopolamine ($n = 10$ mice). **d** Investigation time after the VS increases when mice receive an OB cannula infusion of Ringer's solution, but does not increase when the mice receive scopolamine. Illustration by Ogg, M.C.

($n = 86$ glomeruli from 4 mice; RM ANOVA: $F(2,170) = 82.5$, $p < 0.0001$;   pre-VS $= 23.42 \pm 1.65\%$,   post-VS $= 47.74 \pm 7.58\%$; Fig. 6c, visual stimulation). No increases were seen around these time periods in the control trials (Fig. 6a–c, control).

## Discussion

While previous studies have focused on the mechanisms underlying cortical habituation and dishabituation of prolonged odor input[28–32], relatively little was known about how the OB responds under these conditions. In this study, we characterized excitatory post-synaptic glomerular responses and quantified the amount of habituation that occurs during a continuous odor presentation in both awake and anesthetized mice. On average, glomerular responses rapidly increased with odor onset and then slowly decreased until odor offset. The time course and the relative magnitude of habituation are similar to those measured at both the input[33–35] and output[36–38] levels of the OB during prolonged

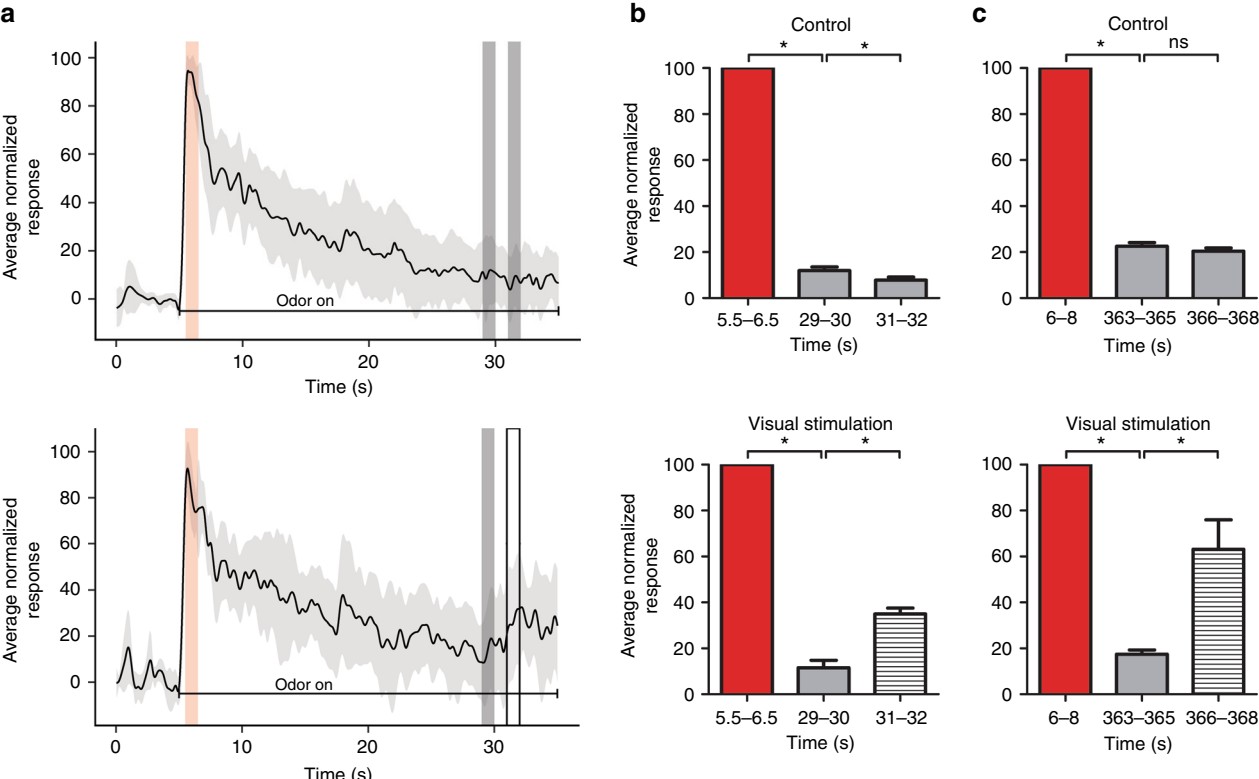

**Fig. 6** A visual context change (VS) dishabituates glomerular odor responses following short and long odor presentations in awake mice. **a** Average fluorescence trace (% of initial response ± s.e.m.) from all recorded glomeruli during control (top trace; $n = 96$ glomeruli, 4 mice) and VS conditions (white box, bottom trace; $n = 73$ glomeruli, 4 mice). **b** During 30 s trials, the average glomerular odor response (% of initial response ± s.e.m.) decreases during the odor presentation and significantly increases following VS. **c** During 7 min trials, the average glomerular odor response (% of initial response ± s.e.m.) decreases during the odor presentation ($n = 92$ glomeruli, 4 mice) and significantly increases following VS ($n = 86$ glomeruli, 4 mice)

odor presentation. Based on this correspondence, and the results from our previous glomerular habituation study[5], it is likely that post-synaptic glomerular habituation is strongly influenced by reduced odor input from the periphery. However, a functional magnetic resonance imaging (fMRI) study in rodents found that while OSN and OB habituation followed identical time courses, OB responses were more reduced, suggesting additional mechanisms within the bulb also partially contribute to OB habituation[35]. In the insect antennae lobe, this effect is mediated by increasing feedback inhibition from interneurons[39–41]. Future studies are needed to determine the extent to which different circuits contribute to OB habituation.

Multiple studies have demonstrated that OB ACh release and receptor activation can increase both glomerular and M/T cell odor responses[17–19]. To test whether this effect can reinstate habituated glomerular odor representations, we electrically stimulated the cholinergic BF near the end of the odor presentation. Our imaging experiment showed that 50 Hz BFS increased the magnitude of glomerular responses, but 5 Hz BFS had no effect. Although direct optogenetic activation of BF cholinergic neuron soma at frequencies as low as 5 Hz can elicit action potentials in brain slices, in vivo stimulation at this low frequency had little to no effect on either inhibitory periglomerular or M/T cell activity in the OB[18].

To determine if this cholinergic effect in the OB could dishabituate odor investigation in awake, behaving animals, we used a newly developed olfactory investigation paradigm coupled with OLS of OB cholinergic fibers in transgenic mice. Based on the frequency-dependent effect seen in our imaging experiment, mice

in the behavioral experiment received OLS at three different frequencies: 5; 25; and 50 Hz. Odor investigation was dishabituated by 25 and 50 Hz OLS. Comparing these two higher-frequency stimulations revealed a frequency dependence of the duration of the dishabituating effect. However, in both cases, in the minute following the stimulation, the amount of dishabituated odor investigation was not significantly different than the amount of initial odor investigation, possibly indicating that if a salient odor is detected, a stereotypical amount of time will be spent investigating that odor, unaffected by the intensity of the stimulation. In our paradigm, 5 Hz OLS had no effect on odor investigation. A study in awake rats found BF firing rates during baseline conditions to be approximately 10 Hz[42]. In the context of these results, our 5 Hz stimulation may be below resting BF firing rates and does not drive cholinergic release above baseline conditions.

These experiments effectively demonstrated that artificially increasing OB ACh can reinstate habituated odor responses and odor investigation, but it was still unclear whether this cholinergic effect in the OB was ecologically valid. Multiple studies have shown that hippocampal and cortical ACh levels rapidly rise in response to various novel sensory stimuli, including contextual changes[22, 23]. Based on this, we performed the behavioral habituation experiment in another group of mice, but instead of driving ACh release optogenetically, we used a more naturalistic paradigm in which the visual context of the chamber was suddenly changed. Visual stimulation increased odor investigation. The effect was not as strong as it was with OLS, however the ACh change that occurs in the OB during the VS is unknown and

could be less intense than that driven by OLS. Alternatively, attention could have been divided between exploring the new visual environment and the reinstated odor. The dishabituating effect of the visual context change was blocked by OB administration of scopolamine, a mAChR antagonist while having no effect on habituation rates themselves. These findings support prior studies demonstrating OB muscarinic activity has little effect on habituation itself[10, 11].

As our initial imaging experiments were on a shorter timescale and in anesthetized mice, it was potentially difficult to directly compare our imaging findings those seen in the behavioral experiments. To address this, we imaged glomerular responses to either 30 s or 7 min odor presentations in awake head-fixed mice. In both cases, responses displayed similar habituation and dishabituation following visual context change, suggesting glomerular habituation/dishabituation mechanisms are similar at either timescale.

Although we did not address the mechanism underlying these effects in this study, they are potentially mediated through muscarinic type 2 receptors located on interneurons within the glomerular layer and external plexiform layer[43, 44]. A subset of these, juxtaglomerular interneurons, tonically inhibit OSNs[45]. This tonic inhibition reduces the gain of OSN excitatory drive onto M/T cells. Following unexpected changes in the environment, we hypothesize that OB ACh release decreases this presynaptic inhibition, increasing the strength of OSN input and enhancing M/T cell responses. In order for an activity-dependent mAChR effect to play a role in our longer-lasting behavioral experiments, input would still have to be coming into the OB from the periphery. Prolonged activity was demonstrated by an fMRI study showing that activity in the olfactory nerve and glomerular layers is reduced, but present even after 10 min of odor presentation[35]. Additionally, none of our cholinergic manipulations were effective at reinstating investigation unless odor was present.

Our experiments showed that ACh can quickly alter glomerular odor responses and shift habituated animals into active investigatory behavior. The BF cholinergic system has been thought to act slowly and globally within the brain[20]. However, more recently it has been shown that fast, local ACh release in specific cortical regions can have a major impact on sensory processing[46] and signal detection[47] on a timescale of seconds. Signal detection studies suggest that phasic cholinergic signaling can arise from local excitation of cholinergic terminals within the cortex, independent of direct BF activation[47–49]. However, given that our behavioral dishabituation effect can be mediated via novel non-olfactory sensory cues, it is likely that the rapid cholinergic dishabituation we observe in the OB is driven by excitatory afferent input into the BF from other brain regions involved in sensory processing or novelty detection. While our effects appear to be localized to the OB, the BF also projects to several other olfactory areas and its activation in these circumstances could certainly affect odor processing as well.

While the BF receives input from many brain regions[50, 51], the locus coeruleus (LC) stands out as a region of interest for future studies. The LC projects noradrenergic fibers to the BF[50, 52–54] and can excite BF cholinergic neurons[55]. LC neurons are activated by novel objects[56] and sensory stimulation[57, 58], such as light flashes, and have been hypothesized to drive arousal-induced attentional processing through BF circuits[59]. Interestingly, the only other report of olfactory dishabituation in the literature involves a noradrenergic mechanism within the piriform cortex[32]. It is possible that the dishabituating cholinergic effects we observe represent an additional pathway by which the LC mediates dishabituation in response to novel environmental stimuli.

In summary, this study revealed a novel dishabituating role for rapid ACh release within the OB, which is necessary and sufficient to change the behavioral salience of sensory input. Here we demonstrated, for the first time, that prolonged odor input leads to decreased responses of excitatory neurons in the glomerular layer and that these responses can be reinstated following brief activation of the cholinergic BF. This effect can be replicated in awake, behaving mice through optogenetic activation of cholinergic fibers in the OB alone. Furthermore, we showed that this effect is ecologically valid as blocking mAChR activation in the OB blocks visual context change-induced dishabituation.

## Methods

**Mice**. Adult male and female mice were used for all experiments. Mice were group-housed and maintained on a 12-h light-dark cycle with ad libitum food and water. All experiments occurred during the light portion of the cycle. All methods were carried out in accordance with relevant and approved guidelines and regulations. All experimental protocols were approved by the University of Tennessee Institutional Animal Care and Use Committee.

**Imaging**. Mice were generated by crossing FVB/N mice expressing Cre recombinase under the Thy1 promoter (FVB/N-Tg(Thy1-cre)1Vln/J; Jax Stock No: 006143) with B6 mice with expressing a cre-dependent green fluorescent $Ca^{2+}$ indicator GCaMP3 (Gt(ROSA)26Sortm38(CAG-GCaMP3)Hze; Jax Stock No: 014538). In the OBs of these mice, GCaMP3 is expressed in excitatory cells. For anesthetized imaging, mice were anesthetized with urethane (2 mg/kg, intraperitoneal (i.p.)) and given an injection of methyl scopolamine (0.05 mg/kg, i.p.) to prevent nasal congestion. Mice were secured in a custom stereotaxic apparatus (Narishige) with a heating pad underneath to maintain body temperature. To create an imaging window, a skin incision was made over the dorsal surface of the mouse head and the bone overlying the OBs was thinned with a dental drill. A bipolar tungsten electrode was stereotaxically implanted in the BF (coordinates: 0.5 mm bregma, 0.6 mm lateral, and ~3.5 mm deep) and fixed to the skull with superglue and dental cement. During imaging sessions, animals were freely breathing and the respiratory rate was monitored from the respiratory oscillation observed in the odor-evoked GCaMP3 odor-evoked signal. For awake imaging, mice were anesthetized with ketamine/xylazine (100/10 mg/kg, i.p.) and given an analgesic injection (carprofen 10 mg/kg, subcutaneous (s.c.)). The bone overlying the dorsal surface of the OB was thinned and covered in cyanoacrylate glue to create a cranial window for optical imaging. An anchor screw was inserted into the parietal bone and a small, stainless steel head bar was attached to the posterior surface of each mouse's skull for head fixation. The head bar and screw were covered in dental cement and mice were allowed two days to fully recover before experimentation. Following recovery, mice were head-fixed on a custom-built treadmill[60, 61] that allowed mice to freely move forward and backward while remaining head-fixed.

Imaging was performed using Scientifica Slicescopes equipped with a ×4 (0.3 numerical aperture) Olympus objective. The dorsal OB was illuminated with a LED light source centered at 480 nm for 40 s/trial. GCaMP signals were band-pass filtered with a Chroma emission filter (HQ535/50) and collected by charge-coupled devise (CCD) cameras at either 25 Hz (30 s trials) or 5 Hz (7 min trials) (NeuroCCD-SM256, Redshirt Imaging). In some anesthetized preparations, the muscarinic antagonist scopolamine hydrobromide (100 μM, dissolved in Ringer's solution; Sigma-Aldrich, USA) was applied to the OB as previously described[17]. For visual dishabituation (VD) imaging experiments, awake mice ($n = 7$) were presented with either a 30-s or a 7-min odor while continuously imaging OB activity. At either 25 s or 6 min, a black and white lined image was projected onto a screen in front of the mouse using a micro projector (Fugetek; FG-957 DLP).

Methyl valerate, benzaldehyde, 3-carene, ethyl butyrate, 2-heptanone, and ethyl valerate (Sigma-Aldrich) were delivered using a flow-dilution olfactometer previously described[2]. Separate flow controllers for the clean air and the pure odorant vapor were used to mix the flow streams at the end of the odor delivery system to achieve an approximate concentration of 0.25, 0.5, or 0.75% saturated vapor (s.v.) at a flow rate of 0.7 L/min. The odor concentration used for each animal was a concentration that activated discrete, stable glomeruli. In cases where there was more than one trial/animal, odor presentations were separated by more than 11 min, based on a previously determined recovery rate[5]. BFS consisted of a 1–3 s, 5 or 50 Hz train (100 μs pulse) with an amplitude range of 30–120 μA delivered 24 s into the odor presentation.

Maps of stimulus-evoked spatial activity and individual glomerular traces were analyzed in R (version 3.3.2). To correct for photobleaching an exponential curve with offset (nonlinear least squares, nls) was fit to the pre and post odor portions of the fluorescence trace extracted from each pixel or region of interest (a discrete, visually identified glomerulus) and then subtracted. Once corrected, traces were smoothed[62]. For awake imaging, recordings were first corrected for moment artifacts using the non-rigid motion correction algorithm (NoRMCorre)[63]. For both the maps and individual glomerular traces, the initial response was measured

by averaging the response during the 25 frames following odor onset for 30 s trials (sampled at 25 fps) or during 5 frames following odor onset for 7 min trials (sampled at 5 fps). Habituation was measured by averaging the response during the 1 s preceding the BFS or VD. Dishabituation was measured by averaging the response from the one second following the BFS or VD.

**Behavior.** 2-methylpyrazine, 2-heptanone, ethyl butyrate, ethyl valerate, isoamyl acetate (100% unless otherwise indicated; Sigma-Aldrich), or air was delivered into a standard mouse open field chamber (OFC; 40 cm $W$ × 40 cm $D$ × 35 cm $H$; Stoelting) through tubing along a top corner. To ensure that mice would not feel the incoming air, they were placed in a standard mouse cage (18.4 cm $W$ × 29.2 cm $D$ × 12.7 cm $H$) with no bedding in the center of the OFC. This method allows odor delivery without human interference or visual cues, which could result in unintended behavioral effects. To prevent odor build-up, a vacuum-pulled air through small holes in the center of the OFC and a HEPA filter was run throughout the experiment. Mice were placed into the apparatus and allowed to acclimate to the environment for ~10 min. Odor investigation, described below, was recorded using a video camera positioned at the side of the OFC and manually scored using ANY-maze (Stoelting) throughout the duration of the trial, including the last 2 min of the acclimation period and the entire odor delivery period. Odor duration lasted 6–9 min, depending on the trial. In cases where there was more than one trial per animal, odor presentations were separated by at least 1 day (Supplementary Fig. 1d, e).

Olfactory investigation was defined as active sampling episodes in which the mouse sniffs with its head lifted above the plane of its body. This included three stereotypical behaviors: head-up; stretched; and reared sniffing (Supplementary Fig. 1a). Pilot studies and detailed observational analysis of mice exploratory behavior in the absence and presence of odor lead us to combine these behaviors into a single metric of olfactory investigation. Behavior was quantified by pressing and holding a defined key in ANY-maze each time olfactory investigatory behavior was observed. The key remained pressed throughout the behavior and released once the behavior was terminated. This allowed us to quantify the time point and length of each investigatory event throughout the trial. Investigatory behavior was not restricted to a specific zone in the testing apparatus due to the tendency of odors to permeate the space. Rates of baseline and odor-evoked active investigation were consistent across different raters, illustrating the ease of behavioral identification.

For optogenetic experiments, mice expressing channelrhodopsin in cholinergic neurons (B6.Cg-Tg(Chat-COP4*H134R/EYFP,Slc18a3)6Gfng/J; Jax Stock No: 014546) and their WT littermates were used. For LED implantation, mice were anesthetized with ketamine (100 mg/kg)/xylazine (10 mg/kg, i.p.) and given a prophylactic injection of carprofen (5 mg/kg, s.c.). Mice were secured in a stereotaxic apparatus (Kopf Instruments) with a heating pad underneath to maintain body temperature. Miniature blue LEDs (Osram; LBW5SN) were soldered to female miniature electrical connectors and implanted over the OBs[21] (Fig. 2b). A skin incision was made over the dorsal surface of the mouse head and the bone overlying the OBs was thinned with a dental drill. The LED was attached over the thinned OBs with superglue. The rest of the skull was covered with black nail polish to prevent light diffusion and the LED was covered and secured with a screw and black dental cement. Mice recovered for at least 1 week before the experiment. Habituation trials were conducted and scored the same as above. Before each trial, the LEDs were plugged into the pulse generator using thin, light-weight wires. OLS, consisted of a 3 s, 5, 25, or 50 Hz train delivered 7 min into the odor presentation. Each mouse ($n = 5$), received each stimulation frequency once across three different days in pseudorandom order. An additional group of three LED mice were also implanted with a thermistor in their nasal cavity[64] to quantify sniffing during dishabituation.

For VD experiments, B6 mice (C57BL/6J; Jax Stock No: 000664) were used. Mice were anesthetized with ketamine (100 mg/kg)/xylazine (10 mg/kg, i.p.) and given a prophylactic injection of carprofen (5 mg/kg, s.c.) for pain control. Mice were secured in a stereotaxic apparatus (Kopf Instruments) with a heating pad underneath to maintain body temperature. Bilateral stainless steel cannula guides (Plastics One; C235GS-5-2.0/SPC) were implanted in the OBs (coordinates: 4.2 mm bregma, 1 mm lateral/each side, and 1 mm deep) and secured with a head screw, superglue, and dental cement. A dummy and a cap (Plastics One; C235DCS-5/SPC and 303DC/1B) covered the cannula guides during the 1 week recovery period. Before each trial began on day 1, mice were randomly assigned to receive 0.5 μl of either vehicle (Ringer's solution) or scopolamine (1 mM, diluted in Ringer's solution; Sigma-Aldrich) bilaterally infused into each OB at a rate of 0.125 μl/min through each cannula using an SP 100i Syringe Pump (World Precision Instruments). Only the following day, the experiment was repeated with each mouse receiving the opposite treatment as the day before. After infusion, the mice were placed in the chamber to acclimate and habituation trials were conducted as above. To drive dishabituation, a micro projector (Fugetek; FG-957 DLP) mounted above the chamber displayed thin, black and white lines on the floor of the cage at minute 7 of odor delivery. This pattern was identical to the one used for imaging experiments. To estimate drug spread, 0.5 μl of dye solution was infused into the cannula in anesthetized mice. Dye spread through all layers, although we cannot rule our preferential effects on different layers. Dye did not extend beyond the bulb.

**Statistics and animal models.** Statistical analyses were performed using Prism software (GraphPad; version 5.03). Assumptions of normality and homogeneity of variance were first assessed using the D'Agostino & Pearson normality test and Bartlett's test. The data passed the majority of these tests and were analyzed using t-test or an appropriate ANOVA with a post hoc Bonferroni multiple comparisons test. Statistical significance was defined as $p \leq 0.05$. All data are reported as mean ± standard error of the mean. Sample sizes were determined based on prior imaging studies and pilot behavioral experiments. Experimenters were not blind to the experimental condition of the animals during data collection.

**Data availability.** The data that support the findings of this study are available from the authors upon reasonable request.

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

## Acknowledgements

This study was supported by NIH grant NIDCD 013779 to M.L.F. and the Pew Biomedical Scholars Program Grant to M.L.F. We thank A. Ogg for help with graphic design.

## Author contributions

All the authors contributed to the research design and paper review. M.C.O., M.B., and J.M.R. collected and analyzed the data. M.C.O. and M.L.F. wrote the paper.

## Additional information

**Competing interests:** The authors declare no competing interests.

