## [Peer Review File · Nature Communications]

Reviewers' comments:

Reviewer #1 (Remarks to the Author):

Here, Ogg et al examine the role of OB acetylcholine on physiological and behavioral habituation to odors. They report that stimulation of the basal forebrain, which contains acetylcholinergic somata, can largely restore the amplitude of habituated M/T cell responses to pre-habituation levels. Next they show that optogenetic activation of Ach fibers in the bulb can restore habituated behavioral investigation. Lastly, they show that pharmacologically antagonizing Ach transmission in the bulb can block behavioral dishabituation by a visual stimulus.

The questions addressed here will interest olfactory scientists as well as the numerous neuroscientists who care about neuromodulation. The experiments are thorough, convincing, and elegant. I gladly endorse this manuscript for publication in Nature Communications. I have two suggestions to make the study even more convincing:

1) Electrical stimulation of the BF will have widespread effects in the brain. Therefore I wonder whether the effects seen in the imaging experiments may be due to indirect polysynaptic paths. For example, BF stimulation may boost activity in olfactory cortices, which feed back to OB, and thus might contribute to the BF stim effect. To allay this concern, I suggest that the authors apply scopolamine to the bulb before odor and BF stimulation. This would reveal the extent to which the dishabituation of glomerular responses is directly due to Ach transmission.

2) I wonder how specific the BF stim effect is with respect to the glomerular activation pattern. The authors give an example in Fig 2b, in which the glomerular pattern post BF-stimulation looks very much like the pre-habituated pattern. It would be stronger if the authors would quantify how similar the pre-habituation maps are to the post-BF stim maps. I recommend the analysis methods found in Soucy et al (2009) to quantify glomerular map similarity.

Reviewer #2 (Remarks to the Author):

Ogg, Bendahmane and Fletcher present interesting and novel results concerning the role of ACh in the olfactory bulb for interrupting a natural habituation process.

First they show that glomerular excitatory processes adapt to an odor over the course of 30 seconds in such a way that calcium signals decrease to about 40% of the original response. They next show that electrical stimulation of basal forebrain can increase a previously adapted response to an odor back to baseline. This is ascribed to cholinergic effects in the OB.

Next they present behavioral results in which mice slowly habituate their behavioral response to an odor over the course of a 6-9 minute odor presentation. This behavioral habituation can be disrupted by optical stimulation of cholinergic afferents in the OB.

Last, the authors show that an abrupt change in visual surroundings can also lead to dishabituation and that this process seems to be dependent on muscarinic cholinergic receptors in the OB.

Individually each of these experiments yields interesting and novel results; however these experiments do not clearly complement each other because of the choices of experimental parameters.

In the first experiments odors are presented for 30 seconds and glomerular activity (presumably merging ET, SA, and MC cells) are recorded. The methods state that respiratory responses can be obtained from the imaging records but these are not reported. Because respiratory signals in awake behaving mice can show odor habituation, this would show to what extent the neural response adaptation shown here has a behavioral component. It is also not clear to what extent OSN

adaptation is reflected here. Electrical stimulation of basal forebrain activates ACh and GABA processes, hence a claim as to the cholinergic nature of this effect cannot be completely supported by these results.

In the second experiment, odors are presented for 6-9 minutes and a behavioral habituation to the odor is observed over the course of this presentation. It is however clearly not observed here over the course of the 30 seconds reported in the imaging results. The behavioral habituation can be reset by optical stimulation of cholinergic afferents to the OB which is a result similar to that observed in imaging. One question arises however that is not discussed: previous experiments by other groups have used means to increase ACh in the OB during behavioral habituation and not reported any effects on odor habituation, only on cross habituation to other odorants.

It is difficult to conclude how these two experiments relate to each other and whether common mechanisms could underlie these because the observations are at such different time scales.

Minor comments:

The authors should cite an experiment by Chaudhury et al recording mitral cell adaptation during repeated odor presentation in the OB as well as an experiment by Shea et al provoking habituation by LC stimulation.

The authors present the imaging data as "M/T cell responses" when they are imaging overall activity in the glomerular layer. Because M/T cell activity is further modulated in deeper layers these imaging data may not capture what is happening with their spiking responses.

Reviewer #3 (Remarks to the Author):

In this manuscript by M. Ogg et al., the authors propose that increased ACh release by electrical basal forebrain stimulation (BFS) and optogenetic activation of cholinergic axonal terminals in the olfactory bulb (OB) are sufficient to change the behavioral salience of olfactory sensory inputs. The authors test the roles of cholinergic modulation by direct (electrical) activation of cholinergic neurons in the basal forebrain, as well as of axonal terminals in the OB in behaving mice. They also monitor changes of GCaMP signals in mitral and tufted cells, the OB output neurons, in anaesthetized mice before and after basal forebrain stimulation.

Although these are interesting experiments, several weaknesses make them difficult to interpret. I have several concerns with interpretation of the results presented here, which, in my opinion, preclude the publication of the manuscript in the current form in Nature Communications.

Major concerns

1) The authors "hypothesized that brief ACh release in the OB could dishabituate glomerular responses during prolonged odors and, as a result, allow the stimuli to be detected and investigated again." at 105 to 107, and further "this study revealed a novel dishabituating role for rapid ACh release at the first synapse with the olfactory pathway" at 464 to 465.

The data presented in the manuscript is however relying on GCaMP signals imaged under widefield setup from the MTCs (mitral/tufted cells), not OSNs. It is important to re-phrase and state the origin of the signals monitored clearly, and avoid referring to them as glomerular responses at the first synapse with the olfactory pathway, given that they are postsynaptic and may be further processed by local circuits in the OB. If the focus is on glomerular activity, then the authors should monitor activity at the presynaptic site – i.e. express GCaMP in the OSNs and perform similar experiments. Importantly, for both MTC (and potential OSNs experiments), it is essential to probe the fluorescence changes before versus upon BFS in the absence of odorants as well (this is missing in the current manuscript) such as to better link it to the behavioral experiments from Figure 4. The authors should also comment on recent studies reporting that the activation of cholinergic neurons changes spontaneous activity of MT (Rothermel et al. 2014, Ma et al. 2012).

Furthermore, given that anaesthesia can have substantial effects on neuronal activity dynamics, to ensure a fair comparison with the behavioral readouts described later in the manuscript, it would be useful to conduct the imaging experiments in awake mice.

2) How does wakefulness change postsynaptic glomerular odor responses?

The authors need to provide more experimental results about postsynaptic glomerular odor responses in awake mice. mAChRs have been involved in the behavioral state-dependent control of dendrodendritic synapses between MCs and GCs (Tsnuno et al. 2008), and wakefulness has been shown to increase responsiveness in GCs (Kato et al. 2012, Cazakoff et al. 2014). Some description of experimental results from anesthetized and awake condition of postsynaptic glomerular odor responses would be very important to understand which different circuits contribute to OB habituation via cholinergic input into the bulb. In the current version of the manuscript, the authors perform pharmacological blocking of mAChRs by implanting a cannula in the OB (1 mm deep from brain surface). This strategy may result in targeting AchR in the granule cell layer which is not mentioned in the manuscript. Interestingly, application of scopolamine does not appear to alter habituation to odors (Figure 5B), which, in my opinion, would be expected, given the proposed role of cholinergic activation by the authors. For example, does the slope of habituation to prolonged odor presentation change in time compared to Ringer injection? Instead, the authors report an effect on the investigation time when the visual context is changed. It remains unclear however what is the connection between blocking Ach signaling in the olfactory bulb and changes in investigation given visual contrast changes in the environment. How much does the mAChR antagonist spread during the behavioral testing time?

The behavioral assays are designed as olfactory investigation assays and not targeted specifically for visual investigation. Can the authors further probe whether visual driven investigation is well reflected by their assays which describe the extent of odor investigation behaviors?

3) In Figure 4D, does the surge in investigation time after optogenetic stimulation of cholinergic axon terminals in the OB depend on local cholinergic activity in the bulb? Or is it a broader effect that may also include contributions emerging, for example, from action potentials fired by BF neuron somata (elicited by antidromic stimulation from the OB projecting axons)? Do BF neurons that project to the OB send axons only to the OB, or to other targets across the brain? For example, recent reports (Rothermel et al. 2014) indicate that cholinergic inputs from basal forebrain to the bulb also extend axons to anterior olfactory nucleus (AON). Are these different from those projecting to the OB? Local application of scopolamine in the bulb may help clarify the relative contributions.

4) Monitoring sniffing. The authors show that optogenetic light stimulation (OLS) alone (in the absence of odor presentation) does not increase investigative behavior (bar graphs, Figure 4C). It would be important to also show the raster data in time, in the same format as in Figure 4D (over a total interval of 11 min), and also quantify potential changes in sniffing after OLS or BF stimulation. Sniffing should also be monitored during the imaging experiments to determine whether an increase in sniff rate could in principle account for the surge in fluorescence.

Minor concerns

- 1) Fig1a. How do the authors correct for bleaching during odor ON period where fluorescence is much higher than in the baseline period? Can bleaching during the odor ON period account in part for the decay in signal?
- 2) Supplementary Figure 1, b. What are the plotted vertical lines? What does the darkness of lines quantify?
- 3) Fig 3c. What happens to habituation to odor in time if an Ach receptor agonist is applied locally in the bulb? Does the slope of habituation become leaner?
- 5) Figure 4 c. blue bar legend should be adjusted. (odor or no odor? – discrepancy with the title of the

panel).

6) At 295, Supplementary Fig 1e-g should be Supplementary Fig 1c-e.

7) mislabeled Figure 5 panels – c vs. d.

We are very grateful to the reviewers for their constructive comments. The reviewers bring up several points that needed to be clarified or further addressed to make our findings more convincing. Based on this input, we have carried out new, additional imaging experiments and clarified several points throughout the manuscript. We think the manuscript has been substantially improved with the implementation of their recommended changes and additions. This revision of the manuscript includes the following major changes:

1. Additional pharmacological experiments demonstrating HDBS-driven glomerular dishabituation can be blocked with a muscarinic antagonist applied to the OB.
2. The inclusion of awake glomerular imaging experiments demonstrating habituation and context change driven dishabituation at both short (30s) and long (7min) timescales.
3. The inclusion of an experiment measuring respiratory rate during OLS-driven behavioral dishabituation demonstrating that OLS does not alter sniff rates.

Point-by-point responses are included below. In the manuscript, reviewer suggested changes have been highlighted with blue font.

Response to Reviewer #1:

Here, Ogg et al examine the role of OB acetylcholine on physiological and behavioral habituation to odors. They report that stimulation of the basal forebrain, which contains acetylcholinergic somata, can largely restore the amplitude of habituated M/T cell responses to pre-habituation levels. Next they show that optogenetic activation of Ach fibers in the bulb can restore habituated behavioral investigation. Lastly, they show that pharmacologically antagonizing Ach transmission in the bulb can block behavioral dishabituation by a visual stimulus.

The questions addressed here will interest olfactory scientists as well as the numerous neuroscientists who care about neuromodulation. The experiments are thorough, convincing, and elegant. I gladly endorse this manuscript for publication in Nature Communications. I have two suggestions to make the study even more convincing:

1) *Electrical stimulation of the BF will have widespread effects in the brain. Therefore I wonder whether the effects seen in the imaging experiments may be due to indirect polysynaptic paths. For example, BF stimulation may boost activity in olfactory cortices, which feedback to OB, and thus might contribute to the BF stim effect. To allay this concern, I suggest that the authors apply scopolamine to the bulb before odor and BF stimulation. This would reveal the extent to which the dishabituation of glomerular responses is directly due to Ach transmission.*

2) *I wonder how specific the BF stim effect is with respect to the glomerular activation pattern. The authors give an example in Fig 2b, in which the glomerular pattern post BF-stimulation looks very much like the pre-habituated pattern. It would be stronger if the authors would*

quantify how similar the pre-habituation maps are to the post-BF stim maps. I recommend the analysis methods found in Soucy et al (2009) to quantify glomerular map similarity.

Response

Prior work from our lab demonstrated that BFS-driven glomerular enhancement was due to OB muscarinic receptor activation, as the effect could be completely blocked with the muscarinic antagonist scopolamine (Bendahmane et al., 2016). We assumed the same process here, although we did not verify. We have added an additional experiment where we compared dishabituation magnitude before and after OB scopolamine to verify the cholinergic effect in the OB. With regard to quantifying similarity, we agree that a comparison of map similarity could be interesting, however, it seems the analysis used in the Soucy paper requires multiple odorants to make comparisons and we only gave a single odorant per dishabituation.

Responses to Reviewer #2:

Ogg, Bendahmane and Fletcher present interesting and novel results concerning the role of ACh in the olfactory bulb for interrupting a natural habituation process. First they show that glomerular excitatory processes adapt to an odor over the course of 30 seconds in such a way that calcium signals decrease to about 40% of the original response. They next show that electrical stimulation of basal forebrain can increase a previously adapted response to an odor back to baseline. This is ascribed to cholinergic effects in the OB.

Next they present behavioral results in which mice slowly habituate their behavioral response to an odor over the course of a 6-9 minute odor presentation. This behavioral habituation can be disrupted by optical stimulation of cholinergic afferents in the OB.

Last, the authors show that an abrupt change in visual surroundings can also lead to dishabituation and that this process seems to be dependent on muscarinic cholinergic receptors in the OB.

Individually each of these experiments yields interesting and novel results; however, these experiments do not clearly complement each other because of the choices of experimental parameters.

1. In the first experiments odors are presented for 30 seconds and glomerular activity (presumably merging ET, SA, and MC cells) are recorded. The methods state that respiratory responses can be obtained from the imaging records but these are not reported. Because respiratory signals in awake behaving mice can show odor habituation, this would show to what extent the neural response adaptation shown here has a behavioral component. It is also not clear to what extent OSN adaptation is reflected here.

Response

All original imaging was done in anesthetized mice in which their respiratory rate remains steady throughout the odor exposure. In this case, there would be no effects of sniff rate on habituation. From our new awake imaging data at both short and long time scales, mice do sometimes display sniff rates that slow over the course of the presentation. We do however see overall reductions in amplitude of the glomerular signal that are similar, albeit slightly stronger, to that seen in anesthetized conditions. The question of how reduced sniff frequency is related to reduced glomerular response amplitude is certainly very interesting, however it is unlikely that our current paradigm can adequately address this question.

The extent to which OSN adaptation is reflected in the postsynaptic signal is unclear, although it likely plays a role. An fMRI study (Schafer et al., 2005) in rats showed similar reduced activity in both the olfactory nerve layer and the glomerular layer after 10 minutes of odor presentation. This study found that while OSN and OB habituation followed identical time courses, OB responses were more reduced, suggesting additional mechanisms within the bulb also partially contribute to OB habituation. This is included in the manuscript.

2. Electrical stimulation of basal forebrain activates ACh and GABA processes, hence a claim as to the cholinergic nature of this effect cannot be completely supported by these results.

Response

Prior work from our lab demonstrated that BFS-driven glomerular enhancement was exclusively due to OB muscarinic receptor activation, as the effect could be completely blocked by the muscarinic antagonist scopolamine directly applied to the OB (Bendahmane et al., 2016). As similar increases were seen here, we assumed the same mechanism was involved, especially given the LED behavioral experiments. However, we agree that we did not fully demonstrate this in the first version. In the revision, we have added an additional experiment where we compared dishabituation magnitude before and after OB scopolamine to verify that the effect is mediated by cholinergic receptors.

3. In the second experiment, odors are presented for 6-9 minutes and a behavioral habituation to the odor is observed over the course of this presentation. It is however clearly not observed here over the course of the 30 seconds reported in the imaging results. The behavioral habituation can be reset by optical stimulation of cholinergic afferents to the OB which is a result similar to that observed in imaging. One question arises however that is not discussed: previous experiments by other groups have used means to increase ACh in the OB during behavioral habituation and not reported any effects on odor habituation, only on cross habituation to other odorants.

Response

Studies manipulating OB AChRs have had little effect on olfactory habituation itself. These studies suggest that OB ACh release is less important for habituation mechanisms and play a bigger role in cross habituation or discrimination of similar odorants. We think our findings fully

support this idea, except in this case ACh increases are important for enhancing OB processing during novel events. We have added this point to the discussion.

4. It is difficult to conclude how these two experiments relate to each other and whether common mechanisms could underlie these because the observations are at such different time scales.

Response

We understand the reviewers concern and originally chose a shorter habituation time to facilitate the imaging experiments. It is certainly clear from our work and others that mice generally take longer to display habituation behaviorally. We chose a much longer time period, in this case several minutes, to ensure the mice were fully habituated before attempting dishabituation. Our assumption was that glomerular responses should be further reduced at six minutes and that increase glomerular responses due to ACh would have similar effects as at thirty seconds. While the behavioral experiments support this, we presented no imaging evidence that this was the case. Following submission of the original manuscript, we began to focus on repeating our imaging findings in awake mice. We now have included awake imaging experiments in which the odor is presented to head fixed mice for either 30 seconds or 7 minutes to demonstrate similar glomerular habituation responses at both time points (Figure 6). Additionally, we have replicated the behavioral dishabituation experiments while imaging glomerular responses in awake mice at both time scales (Figure 6). We feel these results bolster our original findings by demonstrating dishabituation related increases in glomerular odor responses in awake mice as well as demonstrating that habituation/dishabituation affects are similar at both short and long odor exposure timescales. Together with the optogenetic and pharmacological experiments, we feel this provides a strong case that novel changes in the animal's environment drive olfactory dishabituation through ACh-induced enhancement of glomerular odor responses.

Minor comments:

The authors should cite an experiment by Chaudhury et al recording mitral cell adaptation during repeated odor presentation in the OB as well as an experiment by Shea et al provoking habituation by LC stimulation.

These have been included.

The authors present the imaging data as "M/T cell responses" when they are imaging overall activity in the glomerular layer. Because M/T cell activity is further modulated in deeper layers these imaging data may not capture what is happening with their spiking responses.

We have revised the phrase to M/T cell glomerular responses.

Responses to Reviewer #3:

In this manuscript by M. Ogg et al., the authors propose that increased Ach release by electrical basal forebrain stimulation (BFS) and optogenetic activation of cholinergic axonal terminals in the olfactory bulb (OB) are sufficient to change the behavioral salience of olfactory sensory inputs. The authors test the roles of cholinergic modulation by direct (electrical) activation of cholinergic neurons in the basal forebrain, as well as of axonal terminals in the OB in behaving mice. They also monitor changes of GCaMP signals in mitral and tufted cells, the OB output neurons, in anaesthetized mice before and after basal forebrain stimulation.

Although these are interesting experiments, several weaknesses make them difficult to interpret. I have several concerns with interpretation of the results presented here, which, in my opinion, preclude the publication of the manuscript in the current form in Nature Communications.

Major concerns

1) *The authors “hypothesized that brief Ach release in the OB could dishabituate glomerular responses during prolonged odors and, as a result, allow the stimuli to be detected and investigated again.” at 105 to 107, and further “this study revealed a novel dishabituating role for rapid Ach release at the first synapse with the olfactory pathway” at 464 to 465.*

The data presented in the manuscript is however relying on GCaMP signals imaged under widefield setup from the MTCs (mitral/tufted cells), not OSNs. It is important to re-phrase and state the origin of the signals monitored clearly, and avoid referring to them as glomerular responses at the first synapse with the olfactory pathway, given that they are postsynaptic and may be further processed by local circuits in the OB. If the focus is on glomerular activity, then the authors should monitor activity at the presynaptic site – i.e. express GCaMP in the OSNs and perform similar experiments.

Response

We have clarified this point in the manuscript.

Importantly, for both MTC (and potential OSNs experiments), it is essential to probe the fluorescence changes before versus upon BFS in the absence of odorants as well (this is missing in the current manuscript) such as to better link it to the behavioral experiments from Figure 4. The authors should also comment on recent studies reporting that the activation of cholinergic neurons changes spontaneous activity of MT (Rothermel et al. 2014, Ma et al. 2012).

Response

In our last manuscript (Bendahmane et al., 2016), we performed that exact experiment and found no changes in fluorescence of the glomerular signal. This point is now included in the revised manuscript as well as the references mentioned.

Furthermore, given that anaesthesia can have substantial effects on neuronal activity dynamics, to ensure a fair comparison with the behavioral readouts described later in the manuscript, it would be useful to conduct the imaging experiments in awake mice.

Response

We agree. We now have included awake imaging experiments in which the odor is presented to head fixed mice for either 30 seconds or 7 minutes to demonstrate similar glomerular habituation at both time points. We also performed visual dishabituation experiments at both time scales while imaging in these mice. Overall, we found similar increases in glomerular odor responses at both timescales immediately following VD, suggesting that these increased responses may play a role in driving increased investigation.

2) *How does wakefulness change postsynaptic glomerular odor responses?*

The authors need to provide more experimental results about postsynaptic glomerular odor responses in awake mice. mAChRs have been involved in the behavioral state-dependent control of dendrodendritic synapses between MCs and GCs (Tsnuno et al. 2008), and wakefulness has been shown to increase responsiveness in GCs (Kato et al. 2012, Cazakoff et al. 2014). Some description of experimental results from anesthetized and awake condition of postsynaptic glomerular odor responses would be very important to understand which different circuits contribute to OB habituation via cholinergic input into the bulb.

Response

Our new awake imaging experiments allowed us to directly compare postsynaptic glomerular odor responses to that seen in the anesthetized condition. In general, we finding similar responses of habituation and dishabituation between awake and anesthetized conditions. Interestingly, it appears that the magnitude of habituation in awake mice is stronger than that seen under anesthesia. This would suggest that state-dependent mechanisms like the ones mentioned above or others likely contribute to the extent of habituation.

In the current version of the manuscript, the authors perform pharmacological blocking of mAChRs by implanting a cannula in the OB (1 mm deep from brain surface). This strategy may result in targeting AchR in the granule cell layer which is not mentioned in the manuscript.

Response

It is very difficult to know spread of the drug over time. We have injected identical volumes of dye into the OB through the cannula, to estimate drug spread for this and other OB pharmacological studies in our lab. We find that the entire OB is filled with no spread to the rest of the brain, although we cannot be sure of preferential effects on different layers. This is now included in the methods.

Interestingly, application of scopolamine does not appear to alter habituation to odors (Figure 5B), which, in my opinion, would be expected, given the proposed role of cholinergic activation by the authors. For example, does the slope of habituation to prolonged odor presentation

change in time compared to Ringer injection? Instead, the authors report an effect on the investigation time when the visual context is changed. It remains unclear however what is the connection between blocking Ach signaling in the olfactory bulb and changes in investigation given visual contrast changes in the environment. How much does the mAChR antagonist spread during the behavioral testing time?

Response

This true, similar to our findings, Mandairon et al 2006 infused similar concentrations of scopolamine into the OB of rats and found no changes in habituation. This is now pointed out in the discussion. We compared habituation rates for each group by performing ANOVA and post-hoc tests comparing each hab time point to the initial minute of odor investigation. In both cases, we found that the first time point significantly different from the initial minute was minute 3. This is now included out in the manuscript. This suggests that at least at this timescale, there are no differences in habituation between the ringers and scopolamine group. Given prior studies demonstrating increased ACh release in the cortex and hippocampus following a variety of novel stimuli, we hypothesized that novel environmental changes drive ACh release in the OB as well.

This would allow us to mimic our LED experimental findings in a more naturalistic way and also allow us a way to block mAChR activation in the OB to demonstrate the necessity of OB mAChRs in this process. To determine drug spread, we injected the same volume of dye into the OB and find that it remains in the OB and does not spread into rest of the brain. We have added this point the methods.

The behavioral assays are designed as olfactory investigation assays and not targeted specifically for visual investigation. Can the authors further probe whether visual driven investigation is well reflected by their assays which describe the extent of odor investigation behaviors?

Response

We may be misunderstanding the reviewer's question, but we did investigate whether the visual context change itself lead to increased olfactory investigation in absence of odor stimulation. Using the same time frame, we found no increases in olfactory driven investigation following visual context change alone (Fig. 5d).

3) *In Figure 4D, does the surge in investigation time after optogenetic stimulation of cholinergic axon terminals in the OB depend on local cholinergic activity in the bulb? Or is it a broader effect that may also include contributions emerging, for example, from action potentials fired by BF neuron somata (elicited by antidromic stimulation from the OB projecting axons)? Do BF neurons that project to the OB send axons only to the OB, or to other targets across the brain? For example, recent reports (Rothermel et al. 2014) indicate that cholinergic inputs from basal forebrain to the bulb also extend axons to anterior olfactory nucleus (AON). Are these different*

from those projecting to the OB? Local application of scopolamine in the bulb may help clarify the relative contributions.

Response

Prior work from our lab demonstrated that BFS-driven glomerular enhancement was due to direct OB muscarinic receptor activation, as the effect could be completely blocked with OB bath application muscarinic antagonist scopolamine (Bendahmane et al., 2016). We have added an additional experiment in the revised manuscript where we compared dishabituation magnitude before and after local OB scopolamine to verify the cholinergic effect in the OB. We find that as in our previous paper, OB scopolamine reduces the HDBS-driven increases. Further, we have injected identical volumes of dye into the OB through the cannula to estimate drug spread. We find little to no spread to the rest of the brain. Although we feel the OB dishabituation effect is localized to the OB in this case, BF inputs to other olfactory areas could certainly be influencing other olfactory areas as well. This point has been added to the discussion.

4) Monitoring sniffing. The authors show that optogenetic light stimulation (OLS) alone (in the absence of odor presentation) does not increase investigative behavior (bar graphs, Figure 4C). It would be important to also show the raster data in time, in the same format as in Figure 4D (over a total interval of 11 min), and also quantify potential changes in sniffing after OLS or BF stimulation. Sniffing should also be monitored during the imaging experiments to determine whether an increase in sniff rate could in principle account for the surge in fluorescence.

Response

We have added the raster data of the OLS in the absence of odor to Figure 4. To investigate sniffing, we implanted thermistors in three mice with LEDs to allow us to monitor sniffing. We then subjected them to the same hab/dishab paradigm. We compared mean respiratory rates per second centered around the LED on time. Overall there were no differences seen in respiratory rates seen during the LED activation. This data is included as Supplementary Figure 2. We have included this as supplemental figure 2. For imaging, all mice were anesthetized and do not show changes in sniff rates when odors are presented.

Minor concerns

1) Fig1a. How do the authors correct for bleaching during odor ON period where fluorescence is much higher than in the baseline period? Can bleaching during the odor ON period account in part for the decay in signal?

To avoid this issue, we fit the exponential curve through the pre- and post-odor portions of the fluorescence trace. This assumes there is no additional bleaching component during odor-evoked signal, however we cannot rule out that some does occur. This effect should be small as a majority of the bleaching happens early in the signal.

2) Supplementary Figure 1, b. What are the plotted vertical lines? What does the darkness of lines quantify?

The vertical lines mark investigation events. The darkness signifies the region of maximal events or strongest investigation.

3) Fig 3c. What happens to habituation to odor in time if an Ach receptor agonist is applied locally in the bulb? Does the slope of habituation become leaner?

We did not directly measure this during imaging, however we did look at this during behavioral habituation and found no differences in habituation rate, at least at the time scale of minutes.

5) Figure 4 c. blue bar legend should be adjusted. (odor or no odor? – discrepancy with the title of the panel).

Corrected

6) At 295, Supplementary Fig 1e-g should be Supplementary Fig 1c-e.

Corrected

7) mislabeled Figure 5 panels – c vs. d.

Corrected

REVIEWERS' COMMENTS:

Reviewer #1 (Remarks to the Author):

I have no further comments, the manuscript looks good.

Reviewer #2 (Remarks to the Author):

The authors have added several experiments which address my previous comments.

I have only one comment left that needs to be addressed and which I mentioned before. Because, as stated, imaging captures "all excitatory neurons" in the glomerular layer the authors cannot state that they are imaging M/T glomerular responses, because according to their methods other excitatory responses would also be captured and cannot be distinguished from M/T responses. Because there are excitatory interneurons in the glomerular layer it is important not to over-interpret the data or mislead the reader by claiming that M/T responses are recorded.

The discussion about putative mechanisms involving release of pre synaptic inhibition is interesting but contradictory to data showing an increase, not a decrease of interneuron activity by cholinergic activation

Reviewer #3 (Remarks to the Author):

The authors have responded to my comments and questions adequately and have performed additional experiments that further strengthen the manuscript. I don't have any further concerns.

Reviewer #2 (Remarks to the Author):

The authors have added several experiments which address my previous comments. I have only one comment left that needs to be addressed and which I mentioned before. Because, as stated, imaging captures "all excitatory neurons" in the glomerular layer the authors cannot state that they are imaging M/T glomerular responses, because according to their methods other excitatory responses would also be captured and cannot be distinguished from M/T responses. Because there are excitatory interneurons in the glomerular layer it is important not to over-interpret the data or mislead the reader by claiming that M/T responses are recorded. The discussion about putative mechanisms involving release of pre synaptic inhibition is interesting but contradictory to data showing an increase, not a decrease of interneuron activity by cholinergic activation

Response

The vast majority of glutamatergic excitatory cells with soma located in the glomerular layer or that project dendrites into the glomerulus itself are mitral cells and various tufted cell types (external tufted, superficial tufted, and deep/internal tufted cells). While other excitatory cell types potentially exist, they have not been described to our knowledge. Thus, the odor-evoked glomerular signal would be dominated by these cell types. Based on this, we feel our description of the responses as mitral/tufted cell is correct. Further, our referring to them as mitral/tufted responses was in responses to Reviewer 3's point in the original review to emphasize we are not recording sensory neuron input. In the interest of moving forward, we have changed "M/T cell" to "excitatory postsynaptic" in referring to the glomerular signal we recorded.

In terms of the potential mechanism that involves the release of presynaptic inhibition, the predicted decreased response would be seen in as subset of glomerular inhibitory neurons (dopaminergic/GABAergic) that we did not image in our study.